# Head-Space SPME for the Analysis of Organophosphorus Insecticides by Novel Silica IL-Based Fibers in Real Samples

**DOI:** 10.3390/molecules27154688

**Published:** 2022-07-22

**Authors:** Karolina Delińska, Kateryna Yavir, Adam Kloskowski

**Affiliations:** Department of Physical Chemistry, Faculty of Chemistry, Gdansk University of Technology, 80-233 Gdansk, Poland; kateryna.yavir@pg.edu.pl (K.Y.); adam.kloskowski@pg.edu.pl (A.K.)

**Keywords:** SPME fibers, sample preparation techniques, research development, solid-phase microextraction, ionic liquid, organophosphorus insecticide, food preservation

## Abstract

This work demonstrates the suitability of a newly developed ionic liquid (IL)-based silica SPME fiber for the determination of seven organophosphorus insecticides in cucumber and grapefruit samples by headspace solid-phase microextraction (HS-SPME) with a gas chromatography–flame ionization detector (FID). The sol-gel method released four different sorbent coatings, which were obtained based on a silica matrix containing ILs immobilized inside its pores. In order to obtain ionogel fibers, the following ionic liquids were utilized: 1-Butyl-1-methylpyrrolidinium bis(trifluoromethylsulfonyl)imide; Butyltriethyl ammonium bis(trifluoromethylsulfonyl)imide; 1-(2-Methoxyethyl)-3-methylimidazolium bis(trifluoromethylsulfonyl)imide, and 1-Benzyl-3-methylimidazolium bis(trifluoromethylsulfonyl)imide. The developed fibers were applied for the extraction of seven different insecticides from liquid samples. The most important extraction parameters of HS-SPME coupled with the GC-FID method were optimized with a central composite design. The new SPME fiber demonstrated higher selectivity for extracting the analyzed insecticides compared with commercially available fibers. The limit of detection was in the range of 0.01–0.93 μg L^−1^, the coefficients of determination were >0.9830, and 4.8–10.1% repeatability of the method was found. Finally, the obtained ionogel fibers were utilized to determine insecticides in fresh cucumber and grapefruit juices.

## 1. Introduction

The term pesticides covers chemical and biological substances intended to destroy or delay the development of undesirable organisms. These compounds are used mainly to protect crops against pests, fungal diseases, and weeds, combat rodents and insects during food storage, and protect human health [1]. Pesticides can be transferred between different ecosystems. In their initial form or as derivatives, metabolites can penetrate the soil, water, atmosphere, food products, and animal feed, thus posing a threat to living organisms. One of the pesticide groups that has found widespread use in the food and agriculture industry is organophosphorus insecticides. These insecticides are chemicals that are used to kill diverse types of insects [2]. Over the last few decades, the analysis of organophosphorus insecticides has appeared as a subject of significant matter, mainly due to their potential persistence, toxicity, and water solubility [3]. Organophosphorus compounds are widely used for pest control, demonstrating high insecticidal activity [4]. Organophosphates are readily resorbed after oral, dermal, or inhalation routes of exposure [5]. Organophosphorus insecticides have acute toxicity, contributing to the irreversible inhibition of acetylcholinesterase, which is crucial for the functioning of the central nervous system, usually causing respiratory paralysis and death [6]. Therefore, there is a critical need to determine and quantify trace levels of organophosphorus insecticides in food samples [7].

Based on the available research and literature review, it was established that insecticides appear in environmental samples in relatively low concentrations. Due to this, the extraction of insecticides may be a challenging process, requiring somewhat sensitive sample preparation and chromatographic detection techniques. Therefore, the development of sample preparation techniques compatible with the low concentration of pesticides in various complex matrices is of extreme importance. The improvements implemented in sample preparation techniques usually concern the miniaturization, automation, and solvent-free nature of the sample preparation method, consistent with green analytical chemistry principles. Several microextraction techniques proposed to extract different groups of pesticides have been developed in the last three decades. The most commonly used sample preparation techniques for the isolation and/or enrichment of analytes are liquid–liquid extraction (LLE) and solid-phase extraction (SPE) [8,9]. Nevertheless, the necessity concerning the reduction of the available sample preparation time and the amounts of organic solvents needed to perform the extraction of organic pollutants from environmental samples has provided the development of particular novel extraction approaches, including solid-phase microextraction (SPME) [10] and solvent microextraction [11].

SPME is a solvent-free technique that was developed by Arthur and Pawliszyn in 1990 [12]. This technique allows for the simultaneous extraction and pre-concentration of analytes in two ways: directly from an aqueous sample and from the headspace above the sample. The SPME technique is gaining constantly growing popularity as it can be applied in sampling a wide range of analytes, especially from media characterized by a complex matrix composition, e.g., food or environmental samples [13,14,15]. Among the available sample preparation techniques, the SPME technique deserves special attention due to its many advantages: the ease of use, rapidity, the possibility of both in situ and in vivo sample delivery, the ease of automation, and the elimination of toxic solvent application (the “greenness” of SPME). Nonetheless, the SPME technique is also burdened with some disadvantages; for instance, the limited lifetime of single-fiber use, the possibility of the SPME fiber sorption materials’ degradation at high temperatures during thermal desorption, and the restricted choice of commercially available SPME fibers. The latter issue can be addressed to some extent by combining different extraction modes in one procedure. Furthermore, the utilization of the same fiber in subsequent extractions performed directly from the sample and in the HS mode allows for the efficient isolation of analytes with varying volatility [16,17]. However, the customization of the fiber coating concerning the target analytes can be a game changer. Recently, a significant number of scientific studies were focused on developing and applying new sorption materials as stationary phases for SPME fiber coatings. So far, the most commonly reported were molecularly imprinted polymers, ionic liquids (ILs), polymeric ionic liquids (PILs), conductive polymers, nanoparticles of noble metals, and various carbon-based sorbents (graphene and single- and multi-walled carbon nanotubes) [18]. The vast majority of the mentioned SPME coatings work with an adsorption mechanism (even 90%), involving some drawbacks, e.g., the competitive analytes phenomenon narrows the linearity of the extraction. The abovementioned limitations do not occur when liquid-like materials are applied, since the working mechanism involved in these materials concerns the absorption mechanism. Additionally, these materials should also meet the following criteria: pose a negligible vapor pressure, occur in a liquid phase under a wide range of temperatures, and be thermally stable under relatively high temperatures. Thus, substances that fully comply with these requirements are, no doubt, ionic liquids. Furthermore, ILs have the possibility of being combined from different cation and anion pairs, allowing them to obtain the demanded properties, thus allowing them to be called *designer solvents* [19,20].

The concept of the proposed work assumes obtaining a porous solid silica structure with confined ionic liquid (ionogel) on the surface of the SPME fiber. The obtained solid material is characterized by large pore volumes and diameters, allowing for as high ionic liquid loading as possible. The ILs are applied inside the material’s pores through a dip-coating technique. In this work, four types of SPME fibers were prepared based on the silica matrix (K_2_SiO_3_ and formamide) with the confinement of the following ionic liquids: 1-Butyl-1-methylpyrrolidinium bis(trifluoromethylsulfonyl)imide (IL-1); 1-Benzyl-3-methylimidazolium bis(trifluoromethylsulfonyl)imide (IL-2); 1-(2-Methoxyethyl)-3-methylimidazolium bis(trifluoromethylsulfonyl)imide (IL-3); and Butyltriethyl ammonium bis(trifluoromethylsulfonyl)imide (IL-4). The selection of these ILs was confirmed based on their beneficial parameter differences, such as high or moderate viscosity, and desorption temperatures [21,22].

This work explores the practicability of tuning the extraction properties of developed ionogels with different ionic liquids to extract organophosphorus insecticides as target analytes. The developed ionogel fibers were examined by extracting insecticides from aqueous fruit and vegetable samples (grapefruit and cucumber).

## 2. Materials and Methods

### 2.1. Reagent and Materials

The various insecticides, including diazinon, paraoxon-ethyl, phosalone, dimethoate, fenitrothion, chlorfenvinphos, and heptenophos, were purchased from Sigma-Aldrich (Schnelldorf, Germany) (for the purity of the analytes, see Appendix A). The target compounds were selected to investigate the impacts of the polarity and solubility in the water/octanol system on the extraction process using the developed SPME fibers. Additionally, the selected compounds varied significantly in terms of their molecular sizes, which may have had an influence on the kinetics of the extraction process. Quantitatively, the aforementioned properties were determined by the pKa values (in the range 2.6–8.39), Log K_ow_ (octanol/water partition coefficient) (in the range of 0.78–4.38), and molar masses (in the range of 229.30–367.81). Detailed data are provided in Appendix A. A stock standard solution (100 mg L^−1^) of each compound was prepared in acetonitrile. Furthermore, working standard solutions were prepared by diluting the prepared stock solutions with methanol (HPLC grade, Sigma-Aldrich, Poznan, Poland). The solutions were stored at 4–5 °C.

### 2.2. Instrumentation

All analyses were carried out on a gas chromatograph (Agilent Technologies 7890A GC System, Santa Clara, CA, USA) coupled with flame ionization detection (FID). The GC was equipped with a SPB-5 capillary column (30 m × 0.32 mm ID and 0.25 μm film thickness) that was purchased from Sigma-Aldrich (Poznan, Poland). The GC temperature program commenced at 50 °C for 6 min, after which it was increased successively to 300 °C at 15 °C min^−1^, where it was held for 3 min. The injector (in the splitless mode) and detector were maintained at 220 °C and 300 °C, respectively. Furthermore, ultrapure hydrogen (>99.99%) was utilized as the carrier gas at a flow rate of 0.8 mL min^−1^. The flow rates of the selected FID gases, i.e., air and nitrogen, were 400 and 30 mL min^−1^, respectively. The SPME holder for the manual sampling and the commercially available fibers (polydimethylsiloxane (PDMS), 100 μm, and polyacrylate (PA), 85 μm) were purchased from Merck (Warsaw, Poland).

### 2.3. Preparation of SPME Fibers

Ionogel-based SPME fibers were prepared based on a procedure briefly involving four steps: introduction of the hydroxy groups to the glass surface, preparation of the sol solution, sol-gel coating, and thermal treatment. First, the glass fiber was prepared according to the procedure reported in our previous work [23]. The ionogel sol solution was prepared from the formamide as a pore-forming agent and K_2_SiO_3_ as a precursor (1:5; *v*/*v*), and mixed in a plastic Eppendorf tube (10 min of mixing). The resulting solution was injected using a syringe into a polymeric PEEK (polyether ether ketone) tube. Then, the pre-treated glass fiber was inserted into the tube with the solution. To prevent the leakage of the sol solution, the bottom and the top of the tube were sealed by a GC septum, and placed in Eppendorf tubes (to avoid evaporation). Finally, after gelation, the fiber was removed from the PEEK tube and used in further experimental steps.

### 2.4. HS-SPME Procedure

Six types of SPME fibers, including four ionogel fibers (1-Butyl-1-methylpyrrolidinium bis(trifluoromethylsulfonyl)imide; Butyltriethyl ammonium bis(trifluoromethylsulfonyl)imide; 1-(2-Methoxyethyl)-3-methylimidazolium bis(trifluoromethylsulfonyl)imide, and 1-Benzyl-3-methylimidazolium bis(trifluoromethylsulfonyl)imide) and two commercial fibers (PDMS and PA), were utilized in this work. SPME analyses were performed using glass vials (15 mL) containing 12 mL of aqueous standard solution or water sample containing 25% (*w*/*w*) Na_2_SO_4_. A glass-coated stirring bar was placed in the vial at a stirring rate of 1800 rpm, after which the vial was sealed with a Teflon-faced septum screw cap, and the solution was thermostated for 50 min at 65 °C. Thereafter, 10.0 μL of the stock solution was injected into 12 mL of the aqueous solution (final pesticide concentration was equal to 83 ppb) and left for 30 min to achieve thermal equilibrium. The fiber was exposed to HS above the sample solution for 70 min at 65 °C. Finally, it was removed from the vial and inserted into the GC injector port for thermal desorption at 220 °C. The insecticides were then thermally desorbed for 10 min.

The following parameters were optimized: pH, Na_2_SO_4_, equilibration, and extraction time. Regarding the utilized set of insecticides, the pH was optimized within the range of 3–11. The equilibration of insecticides in the aqueous sample was performed in a five-set of times: 20, 35, 50, 65, and 80 min. Further, the selected concentration of Na_2_SO_4_ was between 5% and 25% (*w*/*w*). The extraction time was optimized in the range of 30–110 min.

### 2.5. Real Samples

The fruit (grapefruit) and vegetable (cucumber) samples utilized in this study were purchased from a street market in Gdansk, Poland. At first, the samples were blended, and the obtained mixtures were homogenized with a centrifuge (10 min at 4000 rpm), and then filtered. After that, the obtained juices were diluted with ultrapure water in the following percentage concentrations of fruits and vegetables: 75%, 50%, 25%, and 1%. To all of the solutions, Na_2_SO_4_ was added in order to obtain the optimal value of 25% (*v*/*v*). Then, 12 mL of each examined solution, and a stir bar were placed in the 15 mL vial, the vial was sealed with a cap containing a membrane, and the solution was mixed for 20 min at room temperature. Thereafter, 10.0 μL of the insecticide mixture (100 mg L^−1^) was spiked into the solution with continuous stirring for 30 min. Afterward, the obtained solution was thermostated for 50 min at 65 °C. The extraction was performed under the optimal conditions for the obtained ionogel fibers: time, 70 min; temperature, 65 °C.

## 3. Results and Discussion

### 3.1. Characterization of the SPME Fibers

#### 3.1.1. Optical Microscope

An optical microscope was used to visually evaluate the thickness and regularity of the obtained fiber coating. As shown in Figure 1 (exemplary), it is noticeable that the obtained fiber coating is characterized as having regular shape and thickness, being smooth, and having no visible cracks.

The determination of the single fiber diameter was conducted in regard to 11 measurements along the fiber (every 1 mm, total length 10 mm); at the same time, the repeatability of the fiber-to-fiber preparation was determined based on the measurement of five fibers; in the first case, the irregularity of the coating did not exceed 5% (309 µm ± 15), and the fiber-to-fiber (322 µm ± 29) was 8%.

#### 3.1.2. Scanning Electron Microscopy

The fiber coating’s visual appearance, porosity, and regularity were also investigated by scanning electron microscopy (SEM). As shown in Figure 2, the fiber was characterized by visible pores (clear porosity) and a relatively smooth surface with minimal rough areas on the fiber’s coating. Considering that the immobilization of the ionic liquid in the pores is based on the action of capillary forces, a too-large pore diameter seems an undesirable property. In addition, there were no significant differences in the regularity and visual appearance of the coatings between various ILs utilized for the confinement inside the SPME fiber.

#### 3.1.3. Mercury Injection Capillary Pressure (MICP)

The material of the fiber coating was characterized by the mercury intrusion porosimetry technique. The results of the measurement are shown in Appendix A. In the MICP technique, ranges of measurable pore diameters were used. Based on the collected data, one might notice that the obtained porous material contained almost only pores with diameters in the range of 0.18 μm up to 0.46 μm, with an average of 0.326 μm (Table 1). The pore diameter distribution was slightly asymmetrical with tailing toward the smaller diameters.

#### 3.1.4. Confinement of Ionic Liquid

The aim of this stage was to check the effectiveness of the process of immobilizing the ionic liquid in the pores of the silica material. The procedure was carried out in five stages: evacuating the air from the system, introducing the fiber (for 30 min under vacuum), holding the fiber under vacuum for 2 h until the air was removed from the pores, slow pressure build-up to atmospheric (pressing IL into pores), and washing the fibers with methanol. The effectiveness of the immobilization of the ionic liquid in the material’s pores depended on the efficiency of air removal from the pores and the viscosity of the ionic liquid.

The vacuum during the immobilization of the ionic liquid was obtained from the pump, for which the maximum negative pressure was 10^−2^ mBa. Bearing in mind that the viscosity of the liquid decreases with increasing temperature, in this step, the maximum temperature allowed was used. According to the system used, silicone oil was used to ensure the thermal contact of the ionic liquid with the heating device. The upper limit of the temperatures used was defined when the silicone oil reached the boiling point under applied vacuum conditions. This temperature was about 150 °C; therefore, in the experiments, the temperature of 100 °C was adopted as a safe value.

#### 3.1.5. Thermogravimetric Analysis

The new SPME fiber coatings were tested for the content of immobilized ionic liquid. The evaluation was performed with the use of thermogravimetric analysis (TGA). TGA was performed for the porous silica material containing ionic liquid, and also for the ionic liquid itself (Figure 3 and Appendix A), under the following conditions: heating up to 800 °C (20 °C min^−1^) in an argon atmosphere.

Since the IL was the only thermally unstable component in the studied material, it was possible to calculate the volume of ILs inside the pores of the silica material. Between 350 °C and 480 °C, a mass drop could be observed for the sample containing IL. There was a slight change in the mass drop rate around 490 °C. For comparison, the pure IL was also analyzed by TGA. A similar phenomenon could be noticed in the course of the decomposition of the pure ionic liquid, which can be seen in Appendix A. The estimated mass content of the IL confined in the pores of silica coating was ca. 56%.

### 3.2. Optimization of the HS-SPME Method

#### 3.2.1. Fiber Selection

In the course of the study, four ionic liquids as an extractant were investigated. ILs shares the same bis(trifluoromethylsulfonyl) imide anion, so their extraction abilities depended solely on the cation. Cations differ concerning their polarity, starting from the less polar butyltriethyl ammonium (IL-4) and 1-Butyl-1-methylpyrrolidinium (IL-1) cations containing only C–C sigma bonds in their structures. Ionic liquid IL-3 contained an additional aromatic ring and oxygen heteroatom, while IL-2 contained two aromatic rings. Moreover, commercial fibers with PDMS (polydimethylsiloxane) (100 μm thick) and PA (polyacrylate) (85 μm thick) coatings were used as a reference. PDMS and PA fibers were chosen because the isolation of the analytes was based on the absorption mechanism in both cases. At the same time, due to the different polarities of both coatings, it was possible to assess the significance of this parameter in the extraction of the target insecticides. A comparison of the extraction abilities of the manufactured fibers and a commercial one was performed using the areas of chromatographic peaks determined for the investigated analytes. Based on previous experience, the parameters of the analytical procedure were as follows: 12 mL of 20% aqueous standard Na_2_SO_4_ solution (*w*/*w*) in a 15 mL glass vial, stirring rate of 1800 rpm, and thermostating at 55 °C for 20 min. Thereafter, 10 μL of the stock solution was injected into 12 mL of the aqueous solution and left for 30 min to achieve thermal equilibrium. The fiber was exposed to HS above the sample solution for 35 min at 55 °C. Finally, it was pulled out of the vial and inserted into the GC injector port for thermal desorption at 220 °C. The insecticides were thermally desorbed for 10 min [24].

Considering the differences in the volume of the coating (extractant), the areas were normalized by dividing them by the volume of the extraction phase. As shown in Figure 4, the sum of the peak areas for the fiber with confined ionic liquids based on 1-Benzyl-3-methylimidazolium cation (IL-2) was the highest for all investigated insecticides. The second-best fiber had pores filled with IL-3. The second-most-effective ionic liquid was IL-3, containing the polar 1-(2-Methoxyethyl)-3-methylimidazolium cation for four of the six insecticides used. Only in the case of insecticides containing a benzyl ring with a substituted chlorine atom (chlorfenvinphos and phosalone) were better extraction yields obtained with the use of non-polar fibers, both in terms of the ionic liquids and commercial coatings (PDMS) used. For this reason, the fiber with a coating containing the IL-2 liquid was selected for further research.

#### 3.2.2. Optimization of the HS-SPME Procedure

Ionic liquids based on the bis(trifluoromethylsulfonyl)imide anion are generally characterized by low solubility in water [25]; however, taking into account the ratio of the volume of the ionic liquid to the volume of the aqueous sample, which is about 10^−4^, a significant loss of the ionic liquid from the coating should be expected. This is highly probable under elevated temperature conditions and prolonged exposure of the fiber in the sample. Considering the above, it was assumed that the extraction procedure would be conducted only in the headspace mode. Several factors, such as the extraction temperature, extraction time, equilibration time, pH of the sample, agitation conditions, salting-out effect, and headspace volume, were optimized to obtain the maximum extraction efficiency (EE) of the investigated insecticides by HS-SPME. The stirring rate was set to the maximum of 1800 rpm purposely to accelerate mass transfer into the system. Based on the obtained results of pH optimization, a value of 7 was established as the most desirable (Figure 5D,E). The HS sample volume was kept at the minimum possible (the highest aqueous sample/HS volume ratio) [26], making it possible to expose the fiber at ~1.5 cm from the top of the vial. The obtained results of equilibration time optimization indicated that an equilibration time of 50 min was considered sufficient (Figure 5A–C). The salt effect on the extraction performance is shown in Figure 5F. Regarding the obtained results from the optimization, a 25% (*w*/*w*) Na_2_SO_4_ concentration was chosen as optimum for the salt content in the sample for further experiments. Since the analytes were released from the fiber by the thermal desorption process, the thermal stability of the developed fiber was another parameter that must be considered. Due to this, the developed fibers were exposed to the GC inlet port up to the temperature of 240 °C. As a result, there were no peaks of unknown origin (artifacts) in the blank runs of the analysis performed. The desorption temperature was set at 220 °C, as it was 20 °C lower than the maximum allowable temperature to be used. Under these conditions, the carry-over effects were studied, allowing to set 10 min as the optimum time for the desorption process. To reduce the number of experiments and optimize factors that can influence the extraction efficiency, the design of experiments (DoE) (Appendix A) was utilized. Furthermore, the applied approach also provided the possibility of interdependence amongst the input variables, which were not involved in the “one-variable-at-a-time” methodology. The extraction parameters, i.e., the time and temperature, were optimized by a central composite design (CCD). The plan was developed by random sampling using the Statistica 12 software (StatSoft, Chicago, IL, USA). The CCD plan utilizing different optimized parameters and the corresponding response variables (the sum of the chromatographic peak areas of investigated insecticides) are presented in Appendix A and Figure 5. The significance of the standardized effects of the variables was examined by the analysis of variance (ANOVA) method.

#### 3.2.3. Method Validation

The performances of the HS-SPME-GC-FID method regarding the single fiber and fiber-to-fiber repeatabilities, limits of detection (LOD), limits of quantification (LOQ), and linearity of the calibration curve (R^2^, slope, and intercept) were evaluated under the optimized experimental conditions. The repeatabilities, LODs, and R^2^s of the IL-2-based ionogel fiber are presented in Table 2.

The concentrations of the insecticides ranged from 0.01–83 μg/L. The calibration curves of all of the analytes were linear within the ranges starting at the LOQ level, with the lowest value of 0.9832 obtained for fenitrothion. The LODs of the ionogel fibers were calculated as a signal-to-noise ratio of 3, as specified in Table 2, and were in the ranges of 0.01–0.93 μg/L. The calculated LOQs (signal-to-noise ratio of 9) were in the ranges of 0.03–2.79 μg/L. The repeatability of a single fiber (Table 2) describes the repeatability of the HS-SPME-GC-FID method for the same fiber in five replicates of extraction. The highest relative standard deviation (RSD) of the repeatability was found for fenitrothion and equaled 10.1, which is an acceptable level. The fiber-to-fiber repeatability is a test of the applicability of the proposed procedure to obtain identical ionogel fibers. Here, the obtained RSDs were between 12.1% and 16.4% for diazinon and phosalone, respectively. The performance of each type of the obtained ionogel fiber began to decrease after ca. 60 extraction/desorption cycles. Therefore, a total of 60 cycles was selected as the lifetime of the fiber.

### 3.3. Real Sample Analysis

In the final stage of the evaluation of the developed ionic liquid-based SPME fiber coatings, the prepared fibers were utilized in real sample extractions. The extraction of a set of organophosphorus insecticides was performed from the headspace of water, cucumber, and grapefruit samples.

In the case of samples with a complex chemical composition, such as food samples, the presence of other components may influence the target analytes’ extraction process. The matrix effect was assessed on the basis of the change in the extraction efficiency depending on the degree of juice dilution. Four dilutions were used, with the sample juice contents being 50%, 25%, 10%, and 1% by volume. Additionally, extractions from the headspace of fresh juice samples were performed. The experiments were performed under optimized conditions. The determined dependences of the relative recovery for the investigated insecticides as a function of the dilution of the original juice samples are presented in Figure 6 (grapefruit) and Figure 7 (cucumber). The extraction results from water samples were used as the reference point. As can be seen in both cases, the matrix effect was significant, where the reduction of the extraction efficiency ranged from 20 to 60% and from 20 to 70% for grapefruit and cucumber juice, respectively. In the case of cucumber juice with a juice content of 10%, only for chlorfenvinphos, the relative yield was lower than 90%. At the same time, achieving a similar RR level for the grapefruit sample required a 100-fold dilution of the sample. However, it can be expected that the influence of the matrix mostly concerned the partitioning of the analytes between the sample and the headspace. This assumption may be confirmed by the results published by other authors, where various types of extraction coatings were used [27].

### 3.4. Comparison with Other Pesticide Determination Methods

The HS-SPME-GC-FID method was compared with other previously published methods to determine the target organophosphorus insecticides in water, cucumber, and grapefruit juice samples. Table 3 summarizes the LOD, RSD, R^2^, RR, extraction time, and type of extractant applied for several analytical techniques investigated so far by scientists for the determination of the selected insecticides. Table 3 includes the analytical procedures covering a wide range of extraction techniques, and the final determination methods. Notably, the LODs and RSD for the proposed method were comparable to or better than those of the other methods. Additionally, there was no significant difference in the R^2^ values. Furthermore, the investigation of the compiled procedures using a combination of SPE and DLLME with MS detection allowed for gaining a much lower LOD than described in this work. According to the results, the HS-SPME-GC-FID method is an easy, fast, sensitive, and repeatable method that can be used for the determination of selected insecticides in diverse matrices.

## 4. Conclusions

The HS-SPME-GC-FID method was successfully applied to determine the presence of selected organophosphorus insecticides in fruit and vegetable samples. Four different IL-based fibers were investigated in this work (IL-1, IL-2, IL-3, and IL-4). The IL-based fibers were obtained by the deep-coating technique, which involved the immersion of glass fibers in a mixture of K_2_SiO_3_ and FA, in which the listed ILs were confined separately.

After optimizing the main extraction parameters of each fiber, the sums of the total peak areas were compared. The SPME coating produced with IL-2 exhibited the best extraction efficiency results. Research has shown that the selection of IL can influence the extraction efficiency. Further, by selecting a different IL to be confined, it is possible to conveniently tune the required physicochemical properties of ILs, depending on the class of substances to be extracted. The results indicated that the total sum of the peak areas was higher for the proposed ionogel fibers than that for the commercially available (PDMS and PA). The ionogel coatings achieved promising results for their application as extractants for SPME. Finally, the proposed ionogel fibers were employed to analyze the insecticides in fresh apple and carrot juices. It was revealed that the sample matrix could affect the extraction efficiency of the SPME of insecticides.

## Figures and Tables

**Figure 1 molecules-27-04688-f001:**
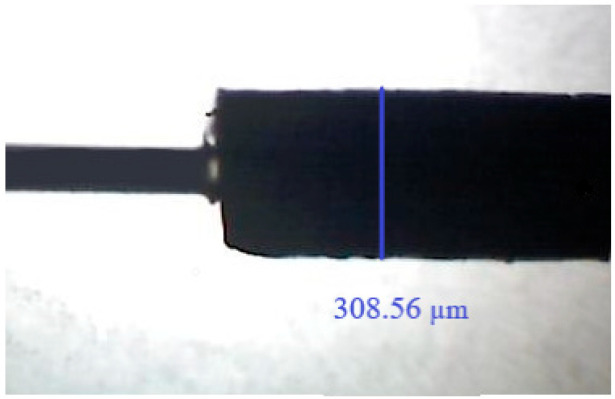
Image of the produced SPME fiber (on the left side, a glass core can be seen) magnified by 50×, and respective diameter.

**Figure 2 molecules-27-04688-f002:**
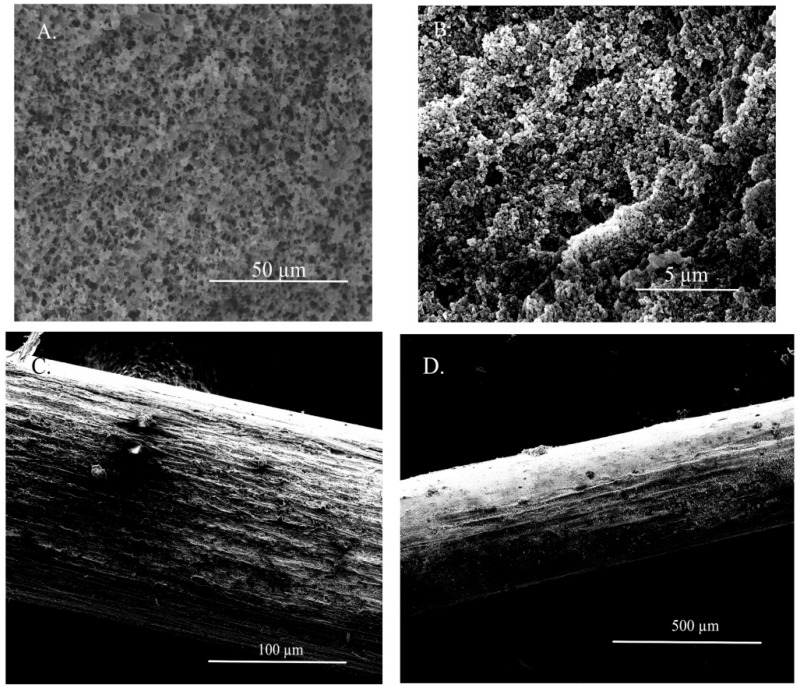
SEM images of the developed SPME fiber in different magnifications (**A**) 50 µm; (**B**) 5 µm; (**C**) 100 µm; (**D**) 500 µm.

**Figure 3 molecules-27-04688-f003:**
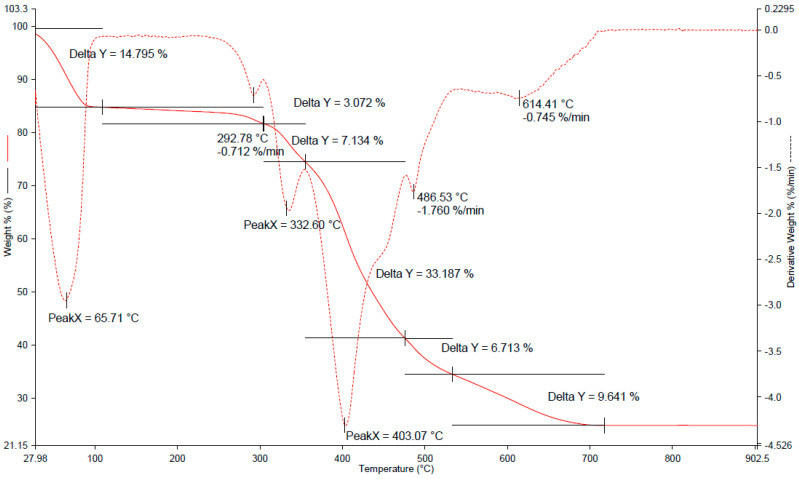
Thermogravimetric curves determined for samples of silica coatings containing ionic liquid IL-2.

**Figure 4 molecules-27-04688-f004:**
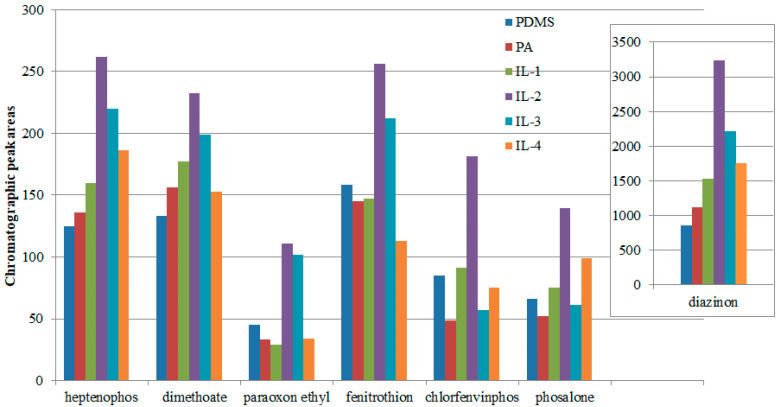
Comparison of the extraction abilities of the investigated insecticides for different ILs, and with reference PA and PDMS commercial fibers, under optimal conditions.

**Figure 5 molecules-27-04688-f005:**
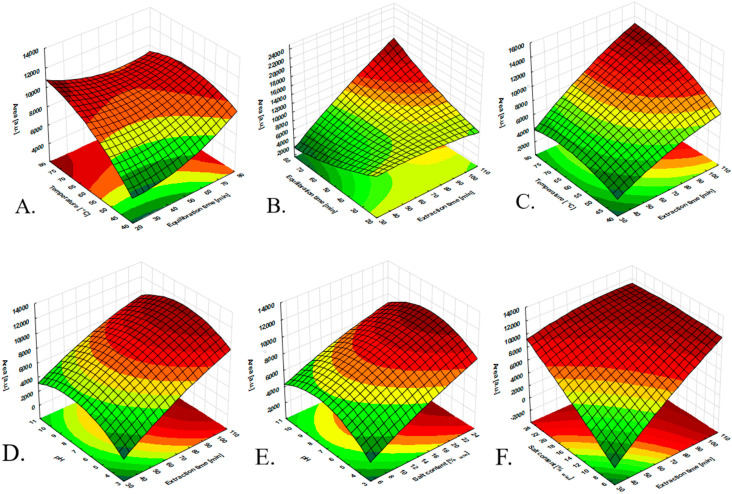
Statistical significance of the effects of the extraction parameters on the extraction performance of the IL-based fiber: (**A**) response surfaces as functions of extraction temperature vs. equilibration time; (**B**) response surfaces as functions of equilibration time vs. extraction time; (**C**) response surfaces as functions of temperature vs. extraction time; (**D**) response surfaces as functions of pH vs. extraction time; (**E**) response surfaces as functions of pH vs. salt content; (**F**) response surfaces as functions of salt content vs. extraction time.

**Figure 6 molecules-27-04688-f006:**
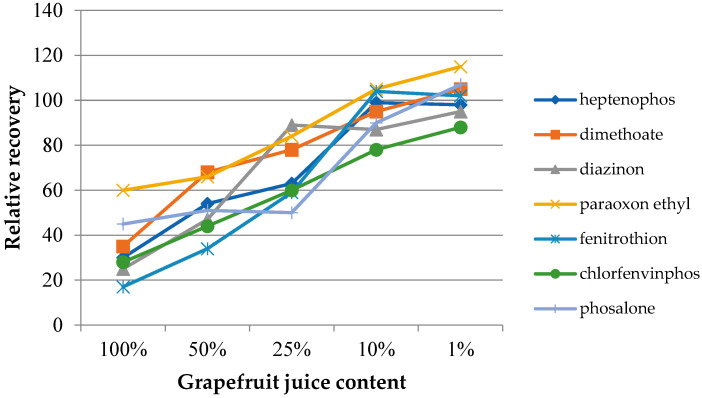
Effect of diluting the samples with ultrapure water at different dilution ratios (*v*/*v*) utilizing the IL-2 fiber for the extraction of grapefruit juice.

**Figure 7 molecules-27-04688-f007:**
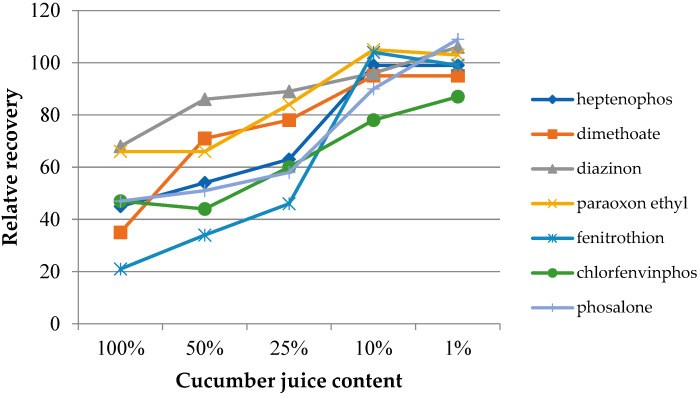
Effect of diluting the samples with ultrapure water at different dilution ratios (*v*/*v*) utilizing the IL-2 fiber for the extraction of cucumber juice.

**Table 1 molecules-27-04688-t001:** Characterization of the obtained porous materials by mercury porosimetry.

Material Evaluated	Median Pore Diameter (Volume) (µm)	Median Pore Diameter (Area) (µm)	Average Pore Diameter (µm)	Total Pore Area (m^2^/g)	Total Porosity (%)
K_2_SiO_3_ + FA	0.362	0.188	0.326	6.33	50.35

**Table 2 molecules-27-04688-t002:** Analytical characteristics of the developed HS-SPME-GC-FID method for organophosphorus insecticide determination using the investigated silica-IL fiber.

Compound	LOD	LOQ	Correlation Coefficient	Single-Fiber Repeatability	Fiber-to-Fiber Repeatability
Heptenophos	0.21	0.63	0.9931	6.4	13.4
Dimethoate	0.50	1.50	0.9925	6.1	14.8
Diazinon	0.01	0.03	0.9956	4.8	12.1
Paraoxon-ethyl	0.60	1.80	0.9850	9.6	12.7
Fenitrothion	0.05	0.15	0.9832	10.1	13.7
Chlorfenvinphos	0.10	0.30	0.9911	7.4	15.5
Phosalone	0.93	2.79	0.9847	8.2	16.4

**Table 3 molecules-27-04688-t003:** Comparison of the proposed HS-SPME-GC-FID method with other methods available in the literature dedicated to the determination of the insecticides investigated in this work.

Analyte	Method	Sample	Real/Standard ^1^	Extraction Phase Used	LOD (µg·L^−1^)	RSD (%)	Extraction Time (min)	Relative Recovery	R^2^	Ref.
Diazinon and fenitrothion	HS-SPME-GC-FID	Apple and carrot juices	Real	Ionogel (Set3, C_4_C_1_Pip, and Set3/C_4_C_1_Pip)	0.01–0.95	2.2–7.4	60	77–114	0.9108–0.9992	[25]
Diazinon	MSPE-DLLME-GC-FID ^2^	Fruits, vegetables, and nectar	Real	Fe_3_O_4_@SiO_2_@ph	0.26	5	2 and 5	100–106	0.9988	[28]
Diazinon	CHLLE-DLLME-GC-FID ^3^	Fruit and vegetable juices	Standard	Di-iso-butyl amine	0.32	6	4	75–99	0.998	[29]
Diazinon, Fenitrothion	MSPE-GC-FID ^4^	Water and fruit juices	Real	KHA/Fe_3_O_4_	0.07–0.14	6.5–8.1	5	89.3–97.3	0.9910–0.9981	[30]
Fenitrothion, Diazinon	SPME-DLLME-GC-MS ^5^	Water, honey, orange, and milk	Standard	C18	0.0005–0.001	0.4–4.71	1	78–98	0.9941–0.9996	[31]
Paraoxon-ethyl	MDCG-MS-DI-SPME ^6^	Peach, orange, and pineapple	Real	PA 65 µm with a layer of PDMS/DVB	n.a.	0.37	60	n.a.	0.9989	[32]
Heptenophos, dimethoate, diazinon, paraoxon ethyl, fenitrothion, chlorfenvinphos, and phosalone	HS-SPME-GC-FID	Water, cucumber, and grapefruit juice	Real	Ionic liquid	0.01–0.93	4.8–10.1	70	85–118	0.9832–0.9956	This work

^1^—The presented LOD values were calculated for real or standard samples (pure water) in the original articles. ^2^—Magnetic solid-phase extraction–dispersive liquid–liquid microextraction–gas chromatography–flame ionization detection. ^3^—Continuous homogenous liquid–liquid extraction–dispersive liquid–liquid microextraction–gas chromatography–flame ionization detection. ^4^—Magnetic solid-phase extraction-gas chromatography–flame ionization detection. ^5^—Solid-phase extraction–dispersive liquid–liquid microextraction–gas chromatography–mass spectrometry. ^6^—Multidimensional gas chromatography-mass spectrometry-direct injection-solid-phase microextraction.

## Data Availability

Not applicable.

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
