# Peer review of "Head-Space SPME for the Analysis of Organophosphorus Insecticides by Novel Silica IL-Based Fibers in Real Samples"

_molecules, 2022, doi:10.3390/molecules27154688_

Round 1

Reviewer 1 Report

After some revisions, this article is well documented and discussed and should be accepted.

In Figure 3, when authors compare the extraction abilities of investigated insecticides for different ILs, the sum of peak areas for fiber with confined ionic liquids is 1-Benzyl-3-methylimidazolium bis(trifluoromethylsulfonyl)imide (IL-2) and not 1-Butyl-1- methylpyrrolidinium bis(trifluoromethylsulfonyl) imide (IL-1).

Also, authors said also that the second-best fiber has pores filled with IL-2 (1-Benzyl-3-methylimidazolium bis(trifluoromethylsulfonyl)imide), but in Figure 3 is observed that is 1-(2-Methoxyethyl)-3-methylimidazolium bis(trifluoromethylsulfonyl) imide (IL-3). 

Author Response

In regard to Reviewer's comment, both issues were changed in the article.

Reviewer 2 Report

The work is interesting, the experiments are generally complete, and the manuscript is generally well written. I have some suggestions and comments that may help to improve it. One generic thing is the poor English in some instances (for example, on line 238 you wrote “The results of the measurement is shown” or line 268 “The another valuable index of the SPME fiber coatings”). Please revise the text to eliminate/reduce these small fails.

Suggestions:

-        If possible, have a native English speaker revising the text.

-        Modify the title of the manuscript to enhance the novelty (development of the new fibre)

-        Change the label of Fig. 1 to “Image of the produced SPME fibre magnified 50 X and respective diameter” instead of “Fabricated SPME fibre with its diameter at 50 times magnification.

Comment 1

In the introduction, you highlight some of the disadvantages of SPME. There are also some strategies to cope with them, as for example, combining DI and HS on the same sample. See for example https://doi.org/10.1016/j.chroma.2013.10.080, and https://doi.org/10.1016/j.chroma.2020.461508, and update accordingly.

Comment 2

Materials and methods section, Lines 131 – 134: please indicate on the text (or table S1) the purity of the reactants.

Comment 3

Materials and methods section, Lines 180 – 181: please identify the supplier of the commercial fibers and justify the selection of the types you used.

Comment 4

Point 2.4: much information that should here is in the section of discussion. Please insert here as much information as possible about experimental conditions (example, ranges evaluated) and limit the info in the discussion to the relevant results.

Comment 5

Did you consider trying DI? Why, why not? Make it clear in the text.

Comment 6

You evaluated repeatability with 5 samples. Consider using more. I suggest that, at least, 8 are used.

Comment 7

You should have also evaluated the inter-day precision. Please do it for at least 5 days.

Comment 8

You used water as the reference to evaluate the recovery of the method. What type of water? Tap? Deionised? Please make it clear.

Comment 9

About the reduced efficiency of extraction of the fiber as function of the reduction of the dilution of the samples (fig. 5 and 6) you wrote “it can be expected that the influence of the matrix mostly concerns the partitioning of the analytes between the sample and the headspace”. To overcome this issue, did you consider using the target matrix to optimize the method (as for example,  https://doi.org/10.1016/j.chroma.2019.06.066)? Why or why not? Was it not possible to find grapefruit or cucumber without the target analytes?

Author Response

The work is interesting, the experiments are generally complete, and the manuscript is generally well written. I have some suggestions and comments that may help to improve it. One generic thing is the poor English in some instances (for example, on line 238 you wrote "The results of the measurement is shown" or line 268 "The another valuable index of the SPME fiber coatings"). Please revise the text to eliminate/reduce these small fails.

Suggestions:

-        If possible, have a native English speaker revising the text.

Regarding the Reviewer's comment, the whole Manuscript was carefully checked in the language context improvement.

-        Modify the title of the manuscript to enhance the novelty (development of the new fibre)

The title was modified: Head-space SPME for the analysis of organophosphorus insecticides by novel silica IL-based fibers in real samples.

-        Change the label of Fig. 1 to "Image of the produced SPME fibre magnified 50 X and respective diameter" instead of "Fabricated SPME fibre with its diameter at 50 times magnification.

 The label of Fig. 1. Was changed regarding the Reviewer's suggestion.

Comment 1

In the introduction, you highlight some of the disadvantages of SPME. There are also some strategies to cope with them, as for example, combining DI and HS on the same sample. See for example https://doi.org/10.1016/j.chroma.2013.10.080, and https://doi.org/10.1016/j.chroma.2020.461508, and update accordingly.

The introduction section was updated accordingly.

Comment 2

Materials and methods section, Lines 131 – 134: please indicate on the text (or table S1) the purity of the reactants.

As it was suggested by the Reviewer, the purity value of all reactants was added to the Supplementary Materials (Table S1).

Comment 3

Materials and methods section, Lines 180 – 181: please identify the supplier of the commercial fibers and justify the selection of the types you used.

The supplier of the commercial fibers is already indicated in the Manuscript. The justification of fibers selection was added to the Manuscript (section 3.2.1. Fiber selection)

Comment 4

Point 2.4: much information that should here is in the section of discussion. Please insert here as much information as possible about experimental conditions (example, ranges evaluated) and limit the info in the discussion to the relevant results.

In regard to the Reviewer's comment, information regarding experimental conditions were moved to section 2.4.

Comment 5

Did you consider trying DI? Why, why not? Make it clear in the text.

The justification for using only headspace extraction is included in section 3.2.2.

Comment 6/ Comment 7

You evaluated repeatability with 5 samples. Consider using more. I suggest that, at least, 8 are used./ You should have also evaluated the inter-day precision. Please do it for at least 5 days.

Increasing the number of repetitions obviously increases the reliability of the results. Nevertheless, the repeatability of the procedure determined using 5 samples is a standard approach in practice. The inter-day precision could not be evaluated due to limited time for revision of the manuscript.

Comment 8

You used water as the reference to evaluate the recovery of the method. What type of water? Tap? Deionised? Please make it clear.

In regard to the Reviewer's comment, there is information in the Manuscript already.

Comment 9

About the reduced efficiency of extraction of the fiber as function of the reduction of the dilution of the samples (fig. 5 and 6) you wrote "it can be expected that the influence of the matrix mostly concerns the partitioning of the analytes between the sample and the headspace". To overcome this issue, did you consider using the target matrix to optimize the method (as for example,  https://doi.org/10.1016/j.chroma.2019.06.066)? Why or why not? Was it not possible to find grapefruit or cucumber without the target analytes?

The manuscript deals with the possibility of using a new porous coating as a carrier for any ionic liquid as an extractant. The study of real samples was aimed at showing the potential of the prepared fibers in analytical practice. The possibility of further in-depth optimization of the insecticide determination method indicated by the reviewer was not our goal. Nevertheless, further research in this direction seems justified based on the results obtained. Regarding the last part of the comment, we would like to emphasize that the tested cucumber and grapefruit samples were spiked with target analytes and initially did not contain them in a concentration that would allow their determination.

Reviewer 3 Report

The manuscript by Delinska et al. describes a novel silica - ionic liquid material system to be applied as  coating on fibers for microextraction in gas chromatographic measurements. The usability of the concept is convincingly demonstrated in the determination of traces of organophosphorus  insecticides in (practically relevant) samples of commercial grapefruit and cucumber.

The paper is interesting and generally well written. The English needs some minor editing to improve legibility and credibility. I will leave this to the editors, but point to the overly generous (and often unnecessary) use of the word "the" throughout the text.

I only have two fairly major points to make concerning the  contents of this study that I would like the authors to attend to:

(1) The description of the interesting new method of preparation of porous silica coatings from silica - formamide system (line 161 and following) is lacking in detail and unclear overall. The text seems to need a little more detail and precision for the reader to follow the essential steps taken. Plus a little more emphasis on Fig 1, e.g. what is seen in the LHS of the image?

(2) To me, the overall (novel!) material properties of the IL-filled porous silica coating remain under-illuminated and nebulous. The authors conclude (line 284) that the amount of IL confined to pores is 56%. Is this weight or volume percentage? Is this independent of the type of IL? Is this 100% of the total available porosity? Etc. This information needs to be provided for readers to judge the characteristics of material and process. There is a real opportunity here to make this more exciting from a physicochemical material properties perspective!

An additional point as to the interaction between Il and porous silica matrix is the capillary filling of the material with IL. Capillary pressure is, amongst other parameters), critically dependent on the contact angle between (specific) IL and silica. What do the authors know about this?

Some minor comments and suggestion, going through the text:

Line 13 Realise

l 24  something not in order with the sentence

l 116 formamide (no capital)

l139 Kow to be defined for general readers

l 161 and following: Too much use of ";" - confusing and unnecessary

3.1.5 I would be in great support of promoting essential graphical TGA information (one Fig) to the main text. It would help the reader to fully appreciate properties of this novel solid-liquid porous system!

l 287 cores

l295 "PA(PDMS)" definition needed

l 315/316 Something wrong? IL2 two times "best"? "pore filled" and "confined" same?

l 414 "matrix effect" needs explanation. What is it?

l 258/259 10-2mBa unbroken on single line

Author Response

The manuscript by Delinska et al. describes a novel silica - ionic liquid material system to be applied as  coating on fibers for microextraction in gas chromatographic measurements. The usability of the concept is convincingly demonstrated in the determination of traces of organophosphorus  insecticides in (practically relevant) samples of commercial grapefruit and cucumber.

The paper is interesting and generally well written. The English needs some minor editing to improve legibility and credibility. I will leave this to the editors, but point to the overly generous (and often unnecessary) use of the word "the" throughout the text.

 I only have two fairly major points to make concerning the  contents of this study that I would like the authors to attend to:

  • The description of the interesting new method of preparation of porous silica coatings from silica - formamide system (line 161 and following) is lacking in detail and unclear overall. The text seems to need a little more detail and precision for the reader to follow the essential steps taken. Plus a little more emphasis on Fig 1, e.g. what is seen in the LHS of the image?

The description of the silica porous coating preparation procedure is deliberately limited to the essential information, such as the steps of synthesis and the list of reagents used. However, for interested readers, we have included a reference to an earlier work where the procedure has been described in detail.

On the LHS in Fig 1. a glass core at which the coating is immobilized might be seen. Since the photo was taken with an optical microscope, its diameter did not match the actual dimensions (the optics were focused on the coating) and its representation in the drawing could be misleading.

(2) To me, the overall (novel!) material properties of the IL-filled porous silica coating remain under-illuminated and nebulous. The authors conclude (line 284) that the amount of IL confined to pores is 56%. Is this weight or volume percentage? Is this independent of the type of IL? Is this 100% of the total available porosity? Etc. This information needs to be provided for readers to judge the characteristics of material and process. There is a real opportunity here to make this more exciting from a physicochemical material properties perspective!

The reviewer's remark is very valuable. Unfortunately, due to the impossibility of determining the density of the silica network, the measured percentage weight loss could not currently be converted into a volume fraction.

An additional point as to the interaction between Il and porous silica matrix is the capillary filling of the material with IL. Capillary pressure is, amongst other parameters), critically dependent on the contact angle between (specific) IL and silica. What do the authors know about this?

The structure of interphase layers of a hybrid material depends on the intermolecular interactions occurring between IL and the material. The surface of silica materials with Si=O and Si-OH groups is likely to form strong interactions with ILs, which leads to IL gathering in the area of its surface. This phenomenon has been proved by the results of many theoretical studies, in which the formation of the multi-layer structure of cations and anions of in-silica confined ILs was observed. This distance range is highly dependent on IL-silica surface interactions competing with forces occurring between IL molecules. However, it may be assumed that a distance shorter than 1 nm is relevant to the subsurface layer. The longer the distance from the surface, the more "bulky" the IL behaviour. In the case of the developed porous material, we are dealing with pores with diameters between 180 nm to 460 nm, which, in the light of the available data, allows us to assume that the immobilized ionic liquid retains its intrinsic properties.

Some minor comments and suggestion, going through the text:

Line 13 Realise

l 24  something not in order with the sentence

l 116 formamide (no capital)

l139 Kow to be defined for general readers

l 161 and following: Too much use of ";" - confusing and unnecessary

3.1.5 I would be in great support of promoting essential graphical TGA information (one Fig) to the main text. It would help the reader to fully appreciate properties of this novel solid-liquid porous system!

l 287 cores

l295 "PA(PDMS)" definition needed

l 315/316 Something wrong? IL2 two times "best"? "pore filled" and "confined" same?

l 414 "matrix effect" needs explanation. What is it?

L 258/259 10-2mBa unbroken on single line

All of the following comments were applied to the Manuscript.